# The Effects of the Combined Co-Expression of GroEL/ES and Trigger Factor Chaperones on Orthopoxvirus Phospholipase F13 Production in *E. coli*

**DOI:** 10.3390/biotech13040057

**Published:** 2024-12-23

**Authors:** Iuliia A. Merkuleva, Vladimir N. Nikitin, Tatyana D. Belaya, Egor. A. Mustaev, Dmitriy N. Shcherbakov

**Affiliations:** 1State Research Center of Virology and Biotechnology VECTOR, Rospotrebnadzor, 630559 Koltsovo, Russia; nikitin_vn@vector.nsc.ru (V.N.N.); belaya_td@vector.nsc.ru (T.D.B.); dnshcherbakov@gmail.com (D.N.S.); 2Department of Natural Sciences, Novosibirsk State University, 630090 Novosibirsk, Russia; e.mustaev@g.ngs.ru; 3Synchrotron Radiation Facility—Siberian Circular Photon Source “SKIF” Boreskov Institute of Catalysis of Siberian Branch of the Russian Academy of Sciences, 630559 Koltsovo, Russia

**Keywords:** orthopoxviruses, monkeypox virus, phospholipase F13, p37, ST-246, NIOCH-14, chaperone, GroEL/ES, trigger factor

## Abstract

Heterologous protein expression often faces significant challenges, particularly when the target protein has posttranslational modifications, is toxic, or is prone to misfolding. These issues can result in low expression levels, aggregation, or even cell death. Such problems are exemplified by the expression of phospholipase p37, a critical target for chemotherapeutic drugs against pathogenic human orthopoxviruses, including monkeypox and smallpox viruses. The complex structure and broad enzymatic activity of phospholipase p37 render it toxic to host cells, necessitating specialized strategies for heterologous expression. In our study, we addressed these challenges using the vaccinia virus F13 protein as a model. We demonstrated that p37 can be effectively synthesized in *E. coli* as a GST fusion protein by co-expressing it with the GroEL/ES chaperone system and Trigger Factor chaperone.

## 1. Introduction

The rapid spread of monkeypox virus in 2022 beyond the African continent has raised significant concerns regarding the global dissemination of zoonotic orthopoxviruses that are pathogenic to humans [1,2,3]. This epidemic highlighted the need to broaden the range of the repertoire of countermeasures against orthopoxvirus infections, as current therapeutic options are limited to a single FDA-approved drug, Tecovirimat (ST-246, TPOXX), and its analog, the prodrug NIOCH-14, which was developed in Russia [4,5].

Tecovirimat specifically targets the membrane protein phospholipase p37 of the enveloped form of orthopoxviruses and exhibits similar half-maximal inhibitory concentration (IC_50_) against vaccinia virus and monkeypox virus [6]. In this study, phospholipase p37 of the vaccinia virus (VACV), referred to as protein F13, was chosen as the model for recombinant p37 protein production.

F13 is a major 42 kDa membrane protein composed of 372 amino acids. In the absence of transmembrane domains, its membrane localization is facilitated by two S-palmitoyl-L-cysteines at positions 185 and 186. This protein is essential for the biogenesis of the bilipid viral membrane and for the egress of the virus from host cells [7]. It exhibits broad enzymatic specificity, including the activities of phospholipase C, phospholipase A, and triacylglycerol lipase, with an additional capacity to exhibit phospholipase D activity [8,9].

Despite the effectiveness of ST-246 therapy, the emergence of resistant strains of monkeypox and vaccinia viruses raises significant concerns, particularly given the lack of alternative specific therapies [4]. Through in silico modeling, a compound with a greater interaction strength than ST-246 was identified; however, it has yet to undergo in vitro or in vivo testing [10]. Since p37 plays a key role in exposing the surface membrane of the enveloped virion, it is considered as a promising target for the development of drugs aimed at treating orthopoxvirus infections [11]. Unfortunately, the search for new p37 inhibitors is complicated by the absence of a crystal structure of the protein, forcing researchers to rely on approximate methods to calculate ligand–protein interaction energies.

Currently, there are a lack of data regarding the development of screening systems based on a functional recombinant variant of p37 or at least extensive experience in its production. The p37 protein can probably be classified as a difficult to express due to its structural characteristics and broad substrate specificity, which encompass multiple enzymatic activities [8]. Therefore, it is essential to devise strategies for the stable and reproducible heterologous expression of functional p37 protein of human pathogenic orthopoxviruses.

*Escherichia coli* is one of the most commonly utilized hosts for protein expression, offering a simple and versatile system suitable for a wide range of proteins. Approaches for the expression of toxic proteins in *E. coli* often include manipulating culture conditions and induction protocols, as well as employing genetic engineering tools. These techniques may involve optimizing expression plasmids, using fusion proteins, the addition of a redox system (GSH/GSSG), and co-expression with chaperones.

Recently, considerable attention has been gained by the role of molecular chaperones in enhancing protein yield in *E. coli*. These proteins facilitate the proper folding and stabilization and maintain the functionality of other proteins within cells [12,13]. The use of chaperones can improve the efficiency of protein synthesis, particularly for proteins with complex secondary and tertiary structures. Chaperones help to prevent the aggregation of misfolded proteins, promoting their correct conformation and increasing the yield of soluble forms of proteins [14]. The primary chaperones involved in this folding process are the ribosome-associated Trigger Factor (TF) and the chaperone systems with their co-chaperones DnaK/DnaJ/GrpE and GroEL/GroES. TF, DnaK/J/GrpE, and GroEL/ES are thought to play distinct yet cooperative roles in protein folding in vivo [15].

In this study, we investigated the impact of two chaperone systems, TF and GroEL/GroES, on enhancing the synthesis of the F13 protein in *E. coli* cells. TF is a ribosome-associated protein and is considered the chaperone that first binds to nascent polypeptide chains [14]. TF binds to the newly synthesized polypeptide chains, stabilizes them and facilitates their transfer to downstream chaperones. Thus, TF can prevent the aggregation and degradation of the F13 protein, thereby enhancing its yield. TF was originally thought to play only a passive role in protein folding. However, it has been proposed that TF works by transferring nascent chains to ATP-dependent chaperone systems such as the DnaKJE and GroEL/ES systems to subsequently assist in folding. Although this is one possible way for TF-associated proteins, there is evidence that TF may participate in protein folding in a more active manner by directly influencing molecule folding [15].

Between the two most promising chaperone systems for correcting misfolded proteins, DnaK/J/GrpE and GroEL/ES, we chose the latter because it interacts with the protein in an isolated system. GroE proteins have been identified in *E. coli* as two proteins, GroEL and GroES, with molecular masses of 57 kDa and 10–15 kDa, respectively [16]. The GroEL chaperone forms an oligomeric structure that encapsulates non-native substrate proteins within a central cavity, which is closed by the GroES co-chaperonin, and corrects misfolded proteins, thereby preventing their stress-induced aggregation. Finally, GroE has been shown to be associated posttranslationally with at least 10–15% of newly synthesized polypeptides in *E. coli*, and it appears that an even larger proportion of misfolded proteins are corrected by this system [14]. We hypothesized that the protein encapsulation mechanism of the GroEL/ES system would be particularly effective for the production of the potentially toxic F13 protein.

The combined co-expression of TF and GroEL/ES chaperones has been successfully applied to produce soluble proteins, including Thermus thermophilus DNA polymerase, human interferon-gamma, ovine growth hormone, and others [13,17,18,19]. Therefore, our objective was to investigate their combined effects on the biosynthesis of the F13 protein in *E. coli*.

## 2. Materials and Methods

### 2.1. Oligonucleotides and Genes

Oligonucleotides, listed in Table 1, were chemically synthesized (Biosset, Novosibirsk region, Russia).

The nucleotide sequence encoding the F13 protein of vaccinia virus was obtained from the GenBank database (accession no. AAY243312.1), codon-optimized and synthesized with a 6× His tag introduced at the C-terminus of the protein in the pUC18-F13 plasmid (DNA-Synthesis, Moscow, Russia).

### 2.2. pET21-GST-F13 Plasmid Construction

The F13L gene was amplified by PCR using the primers gstF13_R and gstF13-3CL _F (Table 1). These primers were designed to introduce a recognition sequence of restriction endonuclease BamHI and the sequence encoding the SARS-CoV-2 3CL protease cleavage site (TSAVLQ↓SG) at the 3′-end of the F13 gene, as well as the HindIII endonuclease site at the 5′-end of the sequence.

Similarly, the GST gene fragment was amplified from the pGEX-2T template using GST_Fau_F and GST-3CL-R primers, which introduced FauNDI (CATATG) and BamHI (GGATCC) (AAGCTT) recognition sequences.

The amplified F13 fragment was subsequently digested with BamHI and HindIII restriction endonucleases, while the GST fragment was cleaved using BamHI and FauNDI. Both DNA fragments were ligated into the pET21a vector, which had been previously cut with FauNDI and HindIII enzymes.

To confirm the correct orientation of the inserts and the integrity of the final pET plasmid structure, Sanger sequencing was performed.

### 2.3. pGroM, pTF, pGTi Plasmid Construction

Plasmids pGroM and pTF were constructed based on the pVAX backbone, which contains the kanamycin resistance gene.

The GroEL/ES chaperone gene was amplified from the *E. coli* strain NEB stable genome using the primers Gro_F and Gro_R (Table 1). PCR was performed according to the standard protocol using PfuSE DNA Pol (SibEnzyme, Novosibirsk, Russia). The temperature profile included initial denaturation at 95 °C for 5 min, 35 amplification cycles (95 °C for 30 s, 63 °C for 15 s, and 72 °C for 4 min) and a final elongation at 72 °C for 5 min. The PCR product was then digested with EcoRI and cloned into pVAX at the EcoRI and EcoRV sites.

Similarly, the TF chaperone gene was amplified from the *E. coli* NEB stable genome using primers TF_F and TF_R (Table 1). The PCR was performed under the same conditions as for GroEL/ES, with an initial denaturation at 95 °C for 5 min, followed by 35 cycles (95 °C for 30 s, 58 °C for 15 s, and 72 °C for 4 min) and a final elongation at 72 °C for 5 min. The resulting PCR product was digested with MfeI and cloned into pVAX at the AfeI and EcoRI sites.

Plasmid pGTi was constructed stepwise from the pVAX plasmid, aiming to integrate chaperone genes, each regulated by the T7 promoter and *lac*-operator. Initially, the region of plasmid pET21a containing *LacI*, *lac*-operator and *T7pro* was amplified using primers Lac_F and Lac_R. This fragment was then inserted into the MluI and FauNDI sites of pVAX, resulting in the intermediate plasmid pVAX-T7, which featured the T7 promoter-*lac* operator system.

Next, the *Lac* operator and *T7pro* were introduced into the GroEL/ES sequence while preserving its own ribosome binding site, using primers GroT7_F and Gro_R. The resulting fragment was digested with ApaI and EcoRI. The TF gene was amplified via PCR using primers TF_Fau_F and TF_R and digested with FauNDI and ApaI.

Both the digested TF and GroEL/ES fragments were then simultaneously cloned into the pVAX-T7 vector at the FauNDI and EcoRI sites

Plasmids pGroM, pTF, and pGTi were amplified in *E. coli* NEB stable cells and sequenced to confirm their structure.

### 2.4. E. coli Cultivation and Protein Expression

*E. coli* BL21 (DE3) cells were heat-shock transformed with the pET21-GST-F13 plasmid, and the transformants were plated on selective LB-agar medium containing ampicillin (100 µg/mL). For co-transformation with pET21-GST-F13 and a chaperone-carrying plasmid (pGroM, pTF, or pGTi), LB-agar was supplemented with a mix of ampicillin (100 µg/mL) and kanamycin (50 µg/mL) for double transformants.

Three to five colonies of *E. coli* from the plates were inoculated into LB liquid medium containing 100 µg/mL ampicillin and 30 µg/mL kanamycin and cultured overnight at 37 °C, 200 rpm.

For protein expression, LB medium supplemented with 0.4% glucose, 100 µg/mL ampicillin and 30 µg/mL kanamycin was used. The overnight culture was diluted 100-fold and cultured at 37 °C, 200 rpm until the optical density reached approximately 0.8–0.9 at 600 nm (OD600). At this point, IPTG was added to a final concentration of 1 mM to induce protein expression. Every 2 h, aliquots of the culture were collected, and cell pellets were harvested by centrifugation at 4500× *g* for 15 min at 4 °C and frozen at −20 °C. The bacterial biomass was resuspended in a lysis buffer (30 mM Tris-HCl, 0.75 M trehalose, 0.5 M NaCl, 2 mM EDTA, 1% triton X-100, 0.5 mg/mL lysozyme, 1 mM PMSF, pH 7.5) and subjected to ultrasonic disintegration. The lysate was centrifuged at 12,000× *g* for 15 min at 4 °C to separate the soluble and insoluble fractions.

The identification of GST-F13 in the protein fractions was performed using SDS-PAGE analysis and Western blotting.

### 2.5. SDS-PAGE Analysis and Western Blotting

An SDS-PAGE analysis of proteins was conducted under denaturing conditions following Laemmli’s method, utilizing 8% polyacrylamide gels. After electrophoresis, the gels were stained with Coomassie Brilliant Blue G-250 to visualize the separated proteins.

For the Western blot analysis, proteins were transferred from the gel to a nitrocellulose membrane using a semi-dry transfer method. Following the transfer, the membrane was blocked with 1% BSA in phosphate-buffered saline (PBS) to prevent the non-specific binding of antibodies. The membrane was then washed three times with PBS containing Tween-20 (PBST). Subsequently, the membrane was incubated for 30 min with a 3000-fold dilution of alkaline phosphatase-conjugated Anti-6X His tag antibody (Abcam, Waltham, MA, USA). After washing, the 1-step BCIP/NBT substrate (ThermoFisher Scientific, Waltham, MA, USA) was added.

### 2.6. GST-F13 Purification and Refolding

Bacterial biomass was harvested by centrifugation at 5000× *g* for 20 min at 4 °C. The resulting bacterial pellet was resuspended in lysis buffer (30 mM Tris-HCl pH 7.5, 0.5 M NaCl, 0.2 mM EDTA, 1% Triton X-100). The suspension was subjected to ultrasonic disintegration 3–5 times at an amplitude of 50% for 1 min. After that, the mixture was centrifuged to separate insoluble and soluble protein fractions. The insoluble protein fraction was then dissolved in a basic buffer for affinity chromatography (30 mM Tris-HCl, 0.5 M NaCl, 8M urea, pH 7.5).

The preparation was then applied to a Ni-NTA column equilibrated with the base buffer. The column was washed five times with five column volumes of the base buffer to remove non-specifically bound proteins. The elution of the bound proteins was performed using a gradient of imidazole (50–500 mM) in the base buffer.

The eluted fractions were analyzed by 8% SDS-PAGE to assess protein purity and yield.

The refolding of GST-F13 was performed using dialysis conducted at 4 °C without stirring. A 2 mL aliquot of the purified protein was placed into a Slide-A-Lyzer MINI Dialysis device (10K MWCO), which was fitted into a 50 mL conical tube. The initial dialysis was carried out against a buffer containing 30 mM Tris-HCl, 2 mM DTT, pH 7.4, 4 M urea. To promote proper refolding, the buffer was changed periodically, with the concentration of urea reduced by half during each exchange. This gradual decrease in urea concentration facilitates the refolding of GST-F13 by allowing the protein folding without inducing aggregation or misfolding. The dialysis process continued until the complete removal of urea was achieved.

The purified protein preparation was stored in stabilizing buffer (30 mM Tris-HCl, 1 M trehalose, 2 mM DTT, 0.01% tween-20, pH 7.4).

### 2.7. p-Nitrophenylphosphorylcholine Assay

The measurement of the phospholipase C activity of F13-GST was carried out by hydrolyzing p-Nitrophenylphosphorylcholine (p-NPPC) (Sigma-Aldrich, St. Louis, MO, USA).

ELISA plates (Corning, Glendale, AZ, USA) were blocked with 1% casein in PBS for one hour at 37 °C. Subsequently, the plates were 3 times washed with reaction buffer (50 mM Tris-HCl, pH 7.2, 10 mM MnCl2, 10 mM CaCl2, and 0.1% Triton X-100).

To each well, 50 µL of 2× reaction buffer, 10 µL of 10 mM aqueous solution of p-NPPC, 30 µL of ddH2O, and 10 µL of GST-F13 solution were added.

The reaction mixture was incubated for 15 min at 37 °C. The absorbance was measured at 405 nm using a Varioskan Lux multimode microplate reader (Thermo Fisher Scientific, Waltham, MA, USA). BSA was used as a negative control.

The enzyme activity was calculated using the following formula:$$A=\frac{∆OD\times Vtotal\times 10^{6}}{Ve\times t\times \varepsilon \times l}$$
where

A—the enzyme activity (U/mL);

∆OD—change in optical density;

Vtotal—total reaction volume (mL);

10^6^—conversion factor from moles to µmol;

Ve—volume of enzyme preparation (mL);

t—incubation time (min);

ε—molar absorption coefficient of p-nitrophenol (18.3 × 10^3^ M^−1^ cm^−1^);

l—cuvette thickness (cm).

The specific activity of the purified enzyme was calculated using the following formula:*SA* = *A*/*C*
where

SA—specific activity of the enzyme (U/mg);

A—enzyme activity (U/mL);

C—enzyme concentration (mg/mL).

The amount of enzyme required to release 1 μmol of p-nitrophenol per minute was taken as a unit of activity.

To test the inhibitory activity, NIOCH-14 was dissolved in DMSO/acetonitrile (1:10) solution and mixed with the enzyme before adding to the reaction mixture.

## 3. Results

### 3.1. Expression of Vaccinia Virus Phospholipase F13 in E. coli

First, we attempted to express F13 as a glutathione-S-transferase (GST) fused protein in *E. coli* BL21 (DE3) cells using the pET21a plasmid system. The GST tag was fused to the N-terminus, while a 6× His-tag was added to the C-terminus of the F13 to facilitate protein purification (Appendix A). Additionally, a cleavage site for coronavirus protease 3CL (TSAVLQ↓SG) was inserted between GST and F13 to enable the removal of GST if necessary by an enzymatic reaction in vitro (Figure 1).

After the induction, we observed a significant slowdown in bacterial growth, and the target protein yield was so low that it was not detectable in the SDS-PAGE analysis (Figure 2A).

The lack of protein induction is likely due to the high toxicity of F13, which is synthesized as a mature protein and exhibits broad substrate specificity, hydrolyzing lipids and phospholipids. In this case, overexpressing the toxic protein as inclusion bodies appears to be an attractive strategy. To ensure overexpression, we constructed the pGroM plasmid, which contains the gene for the GroEL/ES chaperone system, and pTF for the expression of the chaperone Trigger Factor. Both chaperone genes were obtained from the *E. coli* NEB stable genome, including native promoter and terminator regions. These genes were subsequently ligated into the pVAX-based backbone vector carrying the kanamycin resistance gene.

*E. coli* BL21 (DE3) cells were co-transformed with pET21-GST-F13/pGroM and pET21-GST-F13/pTF plasmids and then plated on selective LB-agar containing ampicillin and kanamycin to ensure the selection of double transformants.

GST-F13 expression was induced by adding 1 mM IPTG when the culture reached an OD_600_ of approximately 0.8–0.9. After the induction, bacterial cells were harvested by centrifugation and disrupted using ultrasonic disintegration. The resulting soluble and insoluble protein fractions were analyzed by 8% SDS-PAGE and Western blotting (Figure 2).

The co-expression of GST-F13 (~69 kDa) with each of the chaperones resulted in an increased expression of the target protein in an insoluble form (Figure 2A,B). The highest protein yield was observed after induction at 37 °C; however, reducing the induction temperature to 30 °C and 20 °C led to decreased expression levels and did not yield soluble protein (Figure 2C).

We hypothesized that using a combination of chaperones would have a synergistic effect on protein expression. Additionally, replacing the weak native promoters of the chaperones with strong inducible promoters could facilitate the overexpression of GroEL/ES and TF, further enhancing their impact on GST-F13 expression.

Using the pVAX backbone, we first inserted a region from pET21a that encoded the Lac inhibitor (*LacI*), *lac* operator, and T7 promoter. The TF gene was subsequently placed under the control of this *T7* promoter. The GroEL/ES gene, along with the *lac* operator and *T7pro* sequences added during PCR, was inserted downstream of the TF gene in the plasmid. This resulted in the construction of the pGTi plasmid, which contained both GroEL/ES and TF chaperone genes, each regulated by an inducible T7 promoter (Figure 3).

As expected, the co-expression of GST-F13 with the pGTi plasmid increased protein yield (Figure 4). The overexpression of the protein corresponding to the molecular weight of GroEL (~57 kDa) is clearly evident. However, the presence of the TF protein (~48 kDa) is less visually distinguishable.

A time course induction study of the *E. coli* strain BL21 (DE3)/pET21-GST-F13/pGTi, performed in 50 mL of LB medium, revealed that the peak yield of GST-F13 is observed 6–8 h after IPTG induction (Figure 5).

### 3.2. GST-F13 Purification and Refolding

The GST-F13 protein was isolated from bacterial biomass through a two-step process. In the first stage, ultrasonic disintegration was carried out in a lysis buffer (30 mM Tris-HCl, 0.75 M trehalose, 0.5 M NaCl, 2 mM EDTA, 1% triton X-100, 0.5 mg/mL lysozyme, pH 7.5). The insoluble fraction containing the target protein was then separated by centrifugation at 12,000× *g* for 15 min at 4 °C.

To disrupt inclusion bodies, the pellet was sonicated in the affinity chromatography base buffer (330 mM Tris-HCl, 0.5 M NaCl, pH 7.5) supplemented with 8 M urea. The protein was subsequently purified by affinity chromatography on a Ni-NTA column and eluted with an imidazole gradient. The purity of the preparation was assessed by SDS-PAGE (Figure 6), confirming that GST-F13 corresponds to a predicted molecular mass of approximately ~69 kDa.

Protein refolding was carried out under mild conditions by gradually decreasing urea concentration from 8 M to 0 M. The GST-F13 protein was stored in a stabilizing buffer (30 mM Tris-HCl, 1 M trehalose, 2 mM DTT, 0.01% tween-20, pH 7.4) at 4 °C.

The cultivation of *E. coli* BL21 (DE3)/pET21-GST-F13/pGTi cell cultures in 150 mL shake flasks yielded approximately 50–80 mg of GST-F13 protein per liter of culture medium.

### 3.3. Phospholipase Activity of GST-F13

The phospholipase C activity of the refolded GST-F13 protein was assessed using the chromogenic substrate 4-Nitrophenylphosphorylcholine (pNPPC). The hydrolysis of p-NPPC results in the liberation of p-nitrophenol, a yellow compound that can be quantified by measuring at 405 nm (Figure 7A).

It was confirmed that GST-F13 exhibits phospholipase activity (Figure 7B). Additionally, we investigated the effect of the NIOCH-14 inhibitor on the phospholipase activity of GST-F13. Our results demonstrated that NIOCH-14 solution at a final concentration 0.01 mg/mL reduced the enzyme’s phospholipase activity by at least 50% (Figure 7B).

## 4. Discussion

Heterologous protein expression often presents significant challenges, particularly when the target protein requires posttranslational modifications, is toxic, or is prone to misfolding. These issues can lead to low expression levels, aggregation, or even cell death.

To obtain difficult-to-express proteins, various strategies can be employed, including the selection of an appropriate expression system and the optimization of conditions within commonly used expression systems. This may involve fine-tuning culture conditions, utilizing fusion proteins, and regulating host metabolism. An example of such a protein is phospholipase p37, a critical target for chemotherapeutic drugs against orthopoxvirus infections. Developing a simple, rapid, and accessible method for producing this protein from orthopoxviruses is essential for inhibitor screening, given the significant threat these viruses pose to human health.

A widely used tool for rapid and cost-effective protein production is *E. coli* strain BL21 (DE3). Rapid protein synthesis via the T7 expression system generates high yields of stable protein products [20]. In this study, we also chose *E. coli* BL21 (DE3) because of its lower protease levels, attributed to deficiencies in the Lon and OmpT protease genes, which provides protection for misfolded proteins that are normally targets for degradation.

However, despite the fact that *E. coli* remains the most widely used host for recombinant protein expression, many heterologous proteins are either poorly synthesized in bacterial cells or obtained in nonfunctional forms. Extremely low levels of synthesis (<0.1 mg per 100 mL of culture medium) make subsequent experiments impractical, and one of the most critical challenges in developing heterologous protein expression systems is the lack of adequate protein synthesis [21].

Regarding phospholipase F13, the only published study to date on the production of F13 from the vaccinia virus in *E. coli* (1997) reported a low yield of the GST-F13 fusion protein at just 0.25 mg/L of culture. [8]. The toxicity of enzymatically active proteins often complicates expression, as these proteins can disrupt normal physiological processes in the host, potentially leading to growth inhibition or even cell death. In many cases, this results in extremely low yields of such proteins [21].

To express F13 in *E. coli* BL21 (DE3) cells, we constructed the pET21-GST plasmid, which encodes a polypeptide consisting of GST and F13 separated by the coronavirus protease 3CL cleavage site (Figure 1) for the subsequent removal of GST. A 6× His-tag was added to the C-terminus of the F13 protein to facilitate purification.

Our initial attempts to obtain the GST-F13 protein in *E. coli* BL21 (DE3) cells were unsuccessful (Figure 2A). Although using GST as a fusion partner could promote the proper folding and stabilization of the soluble form of the protein, the protein yield remained low [22].

We hypothesized that a strategy of protein overexpression leading to the formation of aggregates, or so-called “inclusion bodies”, might be effective for the potentially toxic F13 protein. In this scenario, the protein could accumulate in an inactive form within cells and subsequently be purified, with enzymatic activity restored after refolding. To enhance the protein yield, we co-expressed GST-F13 with the chaperones GroEL/ES and TF.

The synergies between these chaperones are complex, and the functional relationship between these chaperone systems is currently only partially understood and deserves further study. Nevertheless, numerous studies have demonstrated the practical benefits of these chaperones to enhance protein production [13,17,18,19].

In this work, we obtained three chaperone plasmids pGroM, pTF, and pGTi. The plasmids pGroM and pTF contained GroEL/ES and TF genes, respectively, while the plasmid pGTi contained both chaperones under the control of the *T7pro* and *lac* operator for the overexpression of the chaperones.

According to the obtained data, the co-expression of GST-F13 with GroEL/ES or TF, each expressed from their native promoters, resulted in an increased yield of the GST-F13 protein yield in inclusion bodies, peaking at an induction temperature of 37 °C (Figure 2). In all instances, a successful induction was achieved at OD600 of approximately 0.8–0.9; earlier or later inductions led to decreased protein yields.

The synthesis of GST-F13 was significantly enhanced by the overexpression of chaperones using the inducible plasmid pGTi (Figure 3). Protein synthesis was observed at 37 °C and 30 °C after 4 h of induction. The immunoblotting results (Figure 4B) using antibodies against His-tag revealed that the protein was present only in the insoluble fractions of bacterial cells.

Additionally, it is noteworthy that the co-expression of GST-F13 simultaneously with GroEL/ES and TF positively influenced the growth characteristics of the culture, promoting the enhanced growth of the producer cells.

The production of GST-F13 in 500 mL flasks containing 150 mL of LB medium took place, with an inoculation dose of 1/100 of the total volume. To bring tighter control over gene expression, 0.4% glucose was added. Induction occurred at an optical density of 0.8–0.9 (OD600) at 37 °C, with shaking at 200 rpm.

GST-F13 was purified on Ni-NTA in the presence of 8 M urea, followed by protein refolding to restore its native structure. The final yield of the GST-F13 was approximately 50–80 mg per liter of culture.

A confirmation of correct refolding of the enzyme is achieved by verifying its functional activity. The p37 protein contains a phospholipase motif (307–334 a.r.) and exhibits phospholipase activity [23,24,25,26]. The motif sequence contains the key fragment HxKxxxxD (where H is histidine, K is lysine, D is aspartic acid, and x is any other a.r.), which is characteristic of phospholipases [25] and consisting of asparagine (N instead of H), lysine, and aspartic acid (Figure 1). Although the structural and functional features of the protein have been described in numerous publications, the geometric parameters of its tertiary structure have not yet been deciphered through experimental methods. Therefore, the tertiary structure of p37 [26,27,28] derived from the folding procedure using the AlphaFold2 methodology is frequently utilized for theoretical studies involving molecular modeling [27,28,29,30,31].

The binding site of the inhibitor Tecovirimat is still controversial. Based on the molecular modeling results, previous studies [27,28,29,30,31,32] have indicated that Tecovirimat binds within the region of the phospholipase domain, forming intermolecular interactions with key amino acid residues, including Asn312 and Lys314.

The tertiary structure model of the GST-F13 protein obtained in this study (Figure 1) showed that the phospholipase domain remains accessible, suggesting that the protein possesses phospholipase activity. This was further confirmed by the hydrolysis of the chromogenic substrate p-nitrophenylphosphorylcholine (p-NPPC), with GST-F13 exhibiting a specific activity of approximately 120–130 U/mg.

While p-NPPC is usually used to measure phospholipase C activity, it can also be hydrolyzed by phospholipase D, albeit to a lesser extent [33]. Among the available substrates, p-NPPC is well suited for measuring p37 activity and can provide an efficient and rapid way for screening a large set of inhibitors. Although a thin-layer chromatography analysis of lipid and phospholipid hydrolysis products can provide a more accurate assessment of the enzyme’s specificity and activity, this approach is labor-intensive and time-consuming, making it unsuitable for high-throughput screening.

The p37 inhibitor NIOCH-14, known as an analog of ST-246 (Tecovirimat), was synthesized at the Novosibirsk Institute of Organic Chemistry in 2009 in collaboration with the State Research Center of Virology and Biotechnology VECTOR (Russia). The authors reported that this compound exhibited high activity in treating orthopoxvirus infections in various animal models [5,34]. We examined the inhibitory effect of NIOCH-14 on the hydrolysis of pNPPC and found that NIOCH-14 significantly inhibited the phospholipase activity of GST-F13. This finding supports the previous hypotheses regarding Tecovirimat’s binding site [27,32] and suggests that our method is suitable for screening potential inhibitors of p37.

Nevertheless, our work has limitations. We made several attempts to express F13 in mammalian cells (CHO-K1 and HEK293T) as well as in the bacterium Bacillus subtilis. Unfortunately, these efforts did not yield the desired protein. It is possible that other potential expression systems, such as yeast, insect cells, or additional mammalian cell lines, as well as cell-free expression systems, may be better suited for synthesizing this protein. Furthermore, our research is limited by the selection of chaperones. We utilized GroEL/GroES and Trigger Factor chaperones; however, exploring alternative or additional chaperones could significantly enhance protein expression efficiency.

## 5. Conclusions

In this study, we successfully expressed and purified the orthopoxvirus F13 protein as a GST fusion protein (GST-F13) in *E. coli* BL21 (DE3) cells by co-expressing it with the chaperones GroEL/ES and TF. Targeting the protein to inclusion bodies proved to be an effective strategy, as it allowed for a higher protein yield and, after refolding, resulted in a functional protein. The purified GST-F13 protein exhibited the expected molecular mass of ~69 kDa and demonstrated phospholipase activity, as confirmed by the hydrolysis of the chromogenic substrate pNPPC.

Additionally, we showed that the NIOCH-14 (analog of ST-246) inhibitor reduced the phospholipase activity of GST-F13. This highlights the potential of using the obtained F13 protein to screen for new phospholipase inhibitors, paving the way for the development of antiviral agents for the specific therapy and prevention of orthopoxvirus infections.

Our results underscore the value of using chaperones to enhance the expression of toxic proteins and provide insights into the inhibitory effects of NIOCH-14 on orthopoxvirus phospholipase activity.

In summary, this study not only emphasizes the importance of optimizing protein expression systems for toxic proteins but also lays the foundation for further research aimed at developing effective therapeutics against orthopoxvirus infections based on p37 inhibitors.

## Figures and Tables

**Figure 1 biotech-13-00057-f001:**
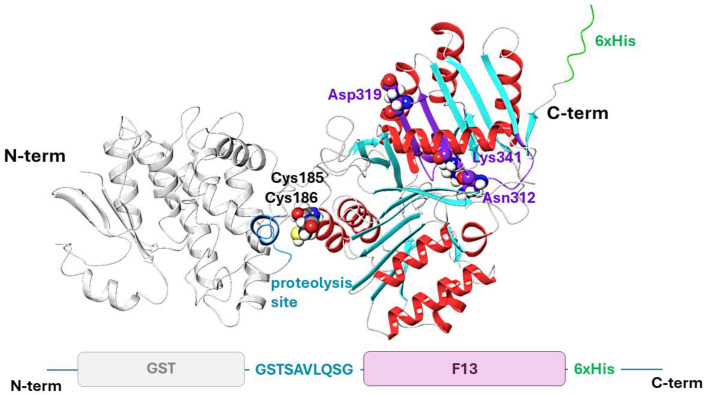
A model of the tertiary structure of the GST-F13 protein (Alphafold2 prediction): The GST protein is represented in light gray, with the 3CL cleavage site highlighted in blue. The secondary structure elements of F13 are color-coded as follows: α-helices are shown in red, β-sheets in blue, and loops in gray. The phospholipase domain is emphasized in purple, as well as the amino acid residues of the phospholipase motif. The non-structural region of the complex, corresponding to the 6× His-tag, is depicted in green.

**Figure 2 biotech-13-00057-f002:**
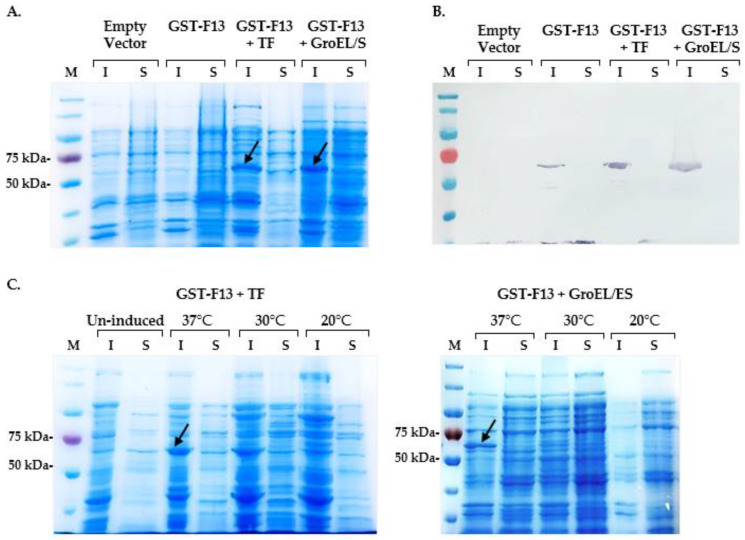
Analysis of extracts of *E. coli* BL21 (DE3) cells co-expressing the GST-F13 protein along with the chaperones Trigger Factor (TF) and GroEL/ES. (**A**) SDS-PAGE of insoluble (I) and soluble (S) protein fractions; (**B**) Western blot analysis of expressed proteins using anti-6× His antibodies; (**C**) SDS-PAGE analysis of insoluble (I) and soluble (S) proteins induced at different temperatures. The GST-F13 protein fraction (~69 kDa) is indicated by arrows.

**Figure 3 biotech-13-00057-f003:**
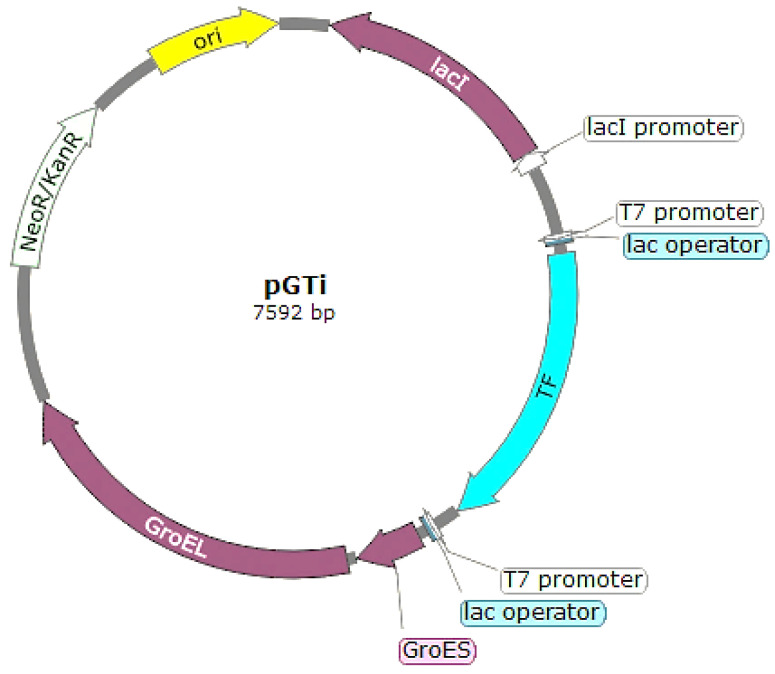
Map of the pGTi plasmid, containing GroEL/ES and Trigger Factor (TF) chaperone genes.

**Figure 4 biotech-13-00057-f004:**
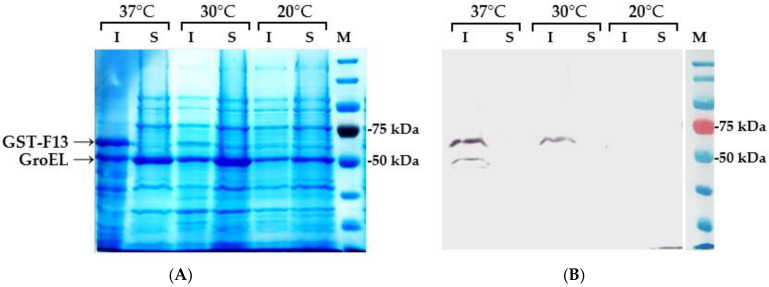
(**A**) SDS-PAGE analysis of insoluble (I) and soluble (S) fractions of *E. coli* BL21 (DE3) cells harboring pET21-GST-F13 and pGTi plasmids, induced with 1 mM IPTG for 4 h; (**B**) Immunoblotting of protein fractions with anti-His antibodies.

**Figure 5 biotech-13-00057-f005:**
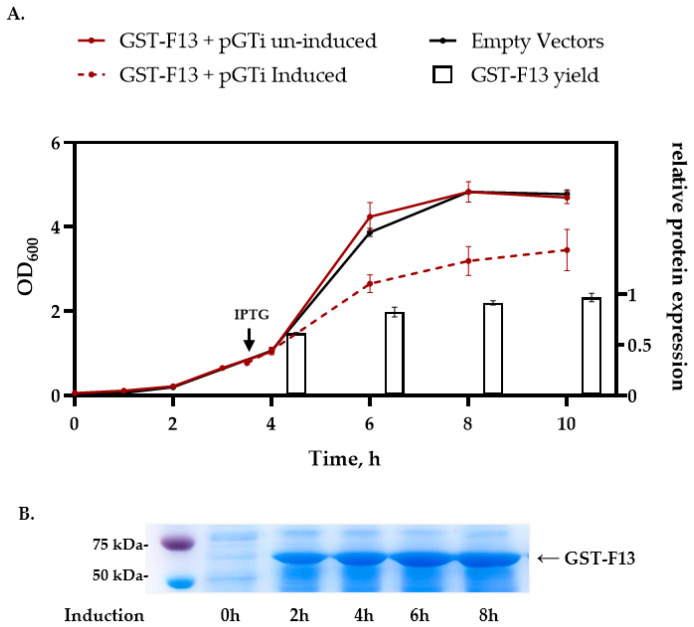
*E. coli* BL21 (DE3)/pET21-GST-F13/pGTi culture growth and GST-F13 protein expression kinetics: (**A**) Growth curve (mean ± SD, *n* = 3) and relative GST-F13 expression. Protein quantity was determined by densitometric analysis of SDS-PAGE (n = 3), and normalized with maximum value. (**B**) SDS-PAGE analysis of the time course production of GST-F13 in *E. coli*.

**Figure 6 biotech-13-00057-f006:**
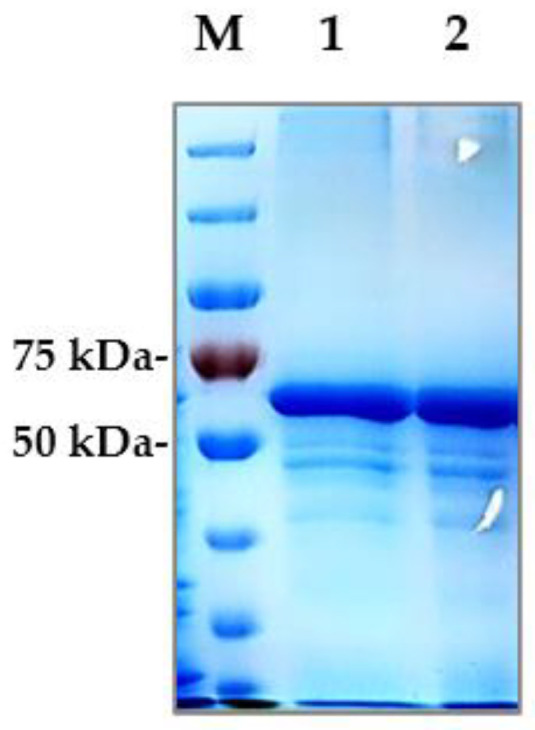
SDS-PAGE analysis of eluted fractions (1, 2) of GST-F13 protein after affinity chromatography.

**Figure 7 biotech-13-00057-f007:**
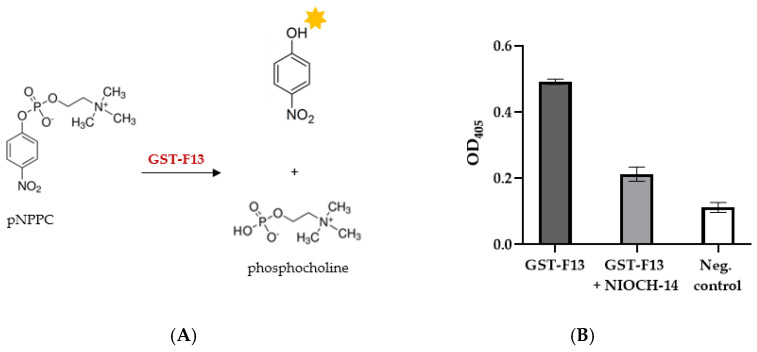
(**A**) Schematic illustration of pNPPC hydrolysis by GST-F13; (**B**) phospholipase activity of GST-F13 protein in the presence of NIOCH-14 inhibitor (0.01 mg/mL). Data are presented as mean ± SD, n = 3.

**Table 1 biotech-13-00057-t001:** Oligonucleotides used for plasmid construction.

Primer	Sequence (5′→3′)
gstF13-3CL_F	aaaaggatccacctcagctgttttgcagagcggtatgtggcctttcgcctctg
gstF13_R	aaaaaagcttagtggtggtgatggtgatgg
GST_Fau_F	aaaaaacatatgtcccctatactaggttatt
GST-3CL_R	aaaaaaggatccttttggaggatggtcgccac
Gro_F	ggcgtcacccataacagatacgg
Gro_R	aaaaagaattccgccactcgcgtcgtccgt
TF_F	aaaaacaattgcttgcggggtaagagttgacc
TF_R	aaaaacgggcccgaaagaggcttttcgcacgg
Lac_F	aaaaaaggcgcgcctgcggcgacgatagtcat
Lac_R	aaaaaagccatatgtatatctccttctt
TF_Fau_F	aaaaacatatgcaagtttcagttgaaacc
GroT7_F	gggcccaaattaatacgactcactataggggaattgtgagcggataacaattccggcgtcacccataacagat

## Data Availability

The original contributions presented in this study are included in the article/Appendix A. Further inquiries can be directed to the corresponding author.

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
