# Peer review of "The Effects of the Combined Co-Expression of GroEL/ES and Trigger Factor Chaperones on Orthopoxvirus Phospholipase F13 Production in E. coli"

_biotech, 2024, doi:10.3390/biotech13040057_

Round 1

Reviewer 1 Report

Comments and Suggestions for Authors

In this article, vaccinia virus F13 can be efficiently synthesized as a GST fusion protein in Escherichia coli by co-expression with the GroEL/ES chaperone system and Trigger Factor chaperone. However, there are still a lot of issues that need to be addressed, and here are the comments I have given.

Q1. The authors describe phospholipase p37 as being post-translationally modified, toxic, or prone to misfolding, so why not try a different expression host such as a yeast with better post-translational modifications?

Q2. Line144-145, When the authors performed expression of GST-F13, they chose to complete the culture 4-5 hours after induction. Why choose to select a shorter induction time when target protein expression is low? How does the growth curve of the strain?

Q3. Line239, Why protein GST-F13 aggregates into insoluble protein fragments only at 37°C. But the GST-F13 protein was not expressed at 30°C and 20°C?

Q4. Line258-261, Co-expression of GST-F13 with the pGTi plasmid, did not achieve soluble expression of GST-F13 protein? In the SDS-PAGE analysis, why there is no band of TF protein, is it not expressed?

Q5. Figure 4B is not clear enough, e.g. protein marker is incomplete

Comments on the Quality of English Language

The manuscript needs lot of language corrections, and I would strongly suggest to the authors to send it to an expert in English editing and academic writing.

Author Response

Dear Reviewer,

Thank you for your prompt review and helpful comments on our manuscript. We have carefully considered your feedback and made significant revisions to enhance the clarity and quality of our work.

We have thoroughly revised the manuscript to address the reviewers' comments and ensure alignment with the specific requirements of the journal BioTech. This revision includes a comprehensive quantitative analysis of the results related to the F13 producer strain, ensuring that all data is presented clearly and accurately. Additionally, we have corrected inaccuracies, errors, and typos throughout the text. We have also added or modified figures to enhance the clarity and understanding of the results. We believe these improvements significantly strengthen the overall quality of the manuscript.

Response to Questions:

Q1. The authors describe phospholipase p37 as being post-translationally modified, toxic, or prone to misfolding, so why not try a different expression host such as a yeast with better post-translational modifications?

  • We made several attempts to express F13 in mammalian cells (CHO-K1 and HEK293T) as well as in the bacterium Bacillus subtilis. Unfortunately, we were unable to obtain the protein. We recognize the limitations of our study and understand the necessity of addressing this in the manuscript. Consequently, we have now expanded the discussion section to mention the potential for using other expression systems, as well as our own experience.

Q2. Line144-145, When the authors performed expression of GST-F13, they chose to complete the culture 4-5 hours after induction. Why choose to select a shorter induction time when target protein expression is low? How does the growth curve of the strain?

  • Unfortunately, we indicated the incorrect induction time. We have made changes to the "Materials and Methods" section and hope that the description of the methods is clear.

To provide further clarity, we have included the growth curve of the producer strain and the SDS-PAGE analysis of protein samples collected at different time points in the revised manuscript. These additions illustrate the relationship between cell growth and protein expression levels. We believe that these enhancements improve the understanding of our experimental design and support our rationale for selecting induction time.

Q3. Line239, Why protein GST-F13 aggregates into insoluble protein fragments only at 37°C. But the GST-F13 protein was not expressed at 30°C and 20°C?

  • We have rephrased the statement for greater accuracy. When the cultivation temperature was decreased, the yield decreased as well, but it was maximized at 37°C.We also added new gel electrophoresis and Western blot analyses to provide a better understanding of the obtained results.

Q4. Line258-261, Co-expression of GST-F13 with the pGTi plasmid, did not achieve soluble expression of GST-F13 protein? In the SDS-PAGE analysis, why there is no band of TF protein, is it not expressed?

  • Indeed, in our experiments, we were unable to obtain soluble protein. Although the expression of chaperones TF and GroEL/ES often enhances protein solubility by stabilizing proteins and correcting misfolded ones, it is possible that protein degradation may occur through mechanisms unrelated to misfolding.

TF is difficult to detect in SDS-PAGE analysis, unlike GroEL, because it is synthesized in low quantities and can be masked by cellular proteins.

Q5. Figure 4B is not clear enough, e.g. protein marker is incomplete

  • We have added a complete marker and revised the figure captions to improve clarity for readers.

Thank you once again for your valuable comments and suggestions. We believe that these revisions have strengthened our manuscript significantly.

Sincerely,

      Iuliia A. Merkuleva (corresponding Author: j.a.merkulyeva@gmail.com) 

Vladimir N. Nikitin

Tatyana D. Belaya,

Egor. A. Mustaev,

Dmitriy N. Shcherbakov

Reviewer 2 Report

Comments and Suggestions for Authors

It is considered that phospholipase p37 is a therapeutic target in vaccinia virus and monkeypox virus infection. Production of recombinant p37 will be useful to conduct screening of inhibitors, however the p37 protein is a challenging target in a bacterial expression system. Here, Merkulev et al. addressed the improvement of the bacterial expression of vaccinia virus p37 protein (F13) by using chaperone system (Trigger factor and GroEL/GroES). Eventually, the chaperone system didn’t help the F13 expression in the soluble fraction, whereas the protein expression level itself was improved by the co-expression of chaperons. The authors recovered the F13 protein from insoluble fraction, and then tested the phospholipase activity in vitro with the refolded F13 protein. I think this kind of study should quantitatively describe the degree of improvement in protein yield. Overall, the authors’ study is qualitative. The contents of the manuscript would be within the category of this journal (BioTech). I will support the publication after my concerns are appropriately addressed.

Major comments:

Page 1, Line 34 – “…orthopoxviruses pathogenic to humans [1].” The authors should cite relevant papers here (e.g., PMID: 35866746 and 36107995).

Page 2, Line 83 – As the authors mentioned in the introduction, there are several chaperones. Please describe why the authors decided to use Trigger factor and GroEL/GroES among the chaperones.

Page 4, Line 171 – The authors utilized the reaction buffer (50 mM HEPES, pH 7.5, 3 mM EGTA, 80 mM KCl, 2.5 mM MgCl2, 1.75 mM CaCl2, and 0.5% Triton X-100) for the assay, but why EGTA which selectively chelate calcium divalent ion was co-excising with CaCl2? Could the authors provide the explanation for this?

Page 5, Line 165 – Protein refolding of is the key part to obtain the functional F13 protein. Could the authors describe the procedure more in detail so that the readership can reproduce your study?

Page 6, Figure 2 – How could the authors specify the arrow-indicated band is GST-F13 without putting the negative control (lysate from E. coli carrying empty plasmid DNA) and/or standard protein? Additionally, the authors should carry out Western blotting to quantitatively compare the relative expression levels upon the changed parameter (with/without TF and temperature).

Page 8, Figure 5 – Could the authors add the yield of GST-F13 protein (mg/L culture) in the main text?

Page 8, Figure 6B – While the authors used BSA as a negative control, His-tagged GST could be the genuine negative control to evaluate the activity of the tagged F13. In addition, it would be more informative if the authors could test the inhibitory activity of NIOCH-14 (IC50).

Page 9, Discussion – The study exclusively focused on GroEL/GroES and Trigger Factor chaperones, whereas exploring alternative or additional chaperones could be resulted in more efficient protein expressions. Also, other protein expression platform (cell-free system, Brevibacillus, yeast, insect cells, and mammalian cells) might be useful to obtain the p37 protein. Could the authors discuss the limitation of this study in the discussion section?

The authors should provide the DNA sequence of plasmid DNA or at least amino acid sequence of GST-F13 as a supplementary information.

Minor Comments:

Page 1, Line 39 – “exhibits the same selectivity index (IC50) for both ...” exhibits similar half-maximal inhibitory concentration (IC50) against vaccinia virus and monkeypox virus.

Page 2, Line 90 – “…virus was obtained from the GenBank database,” Please add gene accession number.

Page 4, Line 148 – “by centrifugation at 4500 × g 15 min,” -> for 15 min,

Page 4, Line 172 – “p37-GST” -> GST-F13

Page 5, Line 205 – “VACV” means what?

Page 5, Line 212 – “After induction, a strong slowdown in the growth of the bacterial culture was observed,” If the authors can present the data, please include it. Otherwise, please add “(data not shown)”.

Page 8, Figure 6B – I suggest the authors add the number of samples (n) and the concentration of NIOCH-14 in the figure legend.

Author Response

Dear Reviewer,

Thank you for your thorough review of our manuscript and your valuable comments. We greatly appreciate your contributions to improving our work.

We have thoroughly revised the manuscript to address the reviewers' comments and meet the specific requirements of the journal BioTech. We incorporated a quantitative description of the results pertaining to the F13 producer strain and corrected any inaccuracies in phrasing and language.

In response to your feedback, we have made the following changes to the manuscript.

Major comments:

Page 1, Line 34 – “…orthopoxviruses pathogenic to humans [1].” The authors should cite relevant papers here (e.g., PMID: 35866746 and 36107995).

  • We have referenced relevant articles. The link to the WHO website has been removed.

Page 2, Line 83 – As the authors mentioned in the introduction, there are several chaperones. Please describe why the authors decided to use Trigger factor and GroEL/GroES among the chaperones.

  • In the introduction, we have provided a justification for our chosen of the chaperones Trigger factor and GroEL/GroES. Additionally, we have relocated the brief description of these chaperones from the discussion section to the introduction.

Page 4, Line 171 – The authors utilized the reaction buffer (50 mM HEPES, pH 7.5, 3 mM EGTA, 80 mM KCl, 2.5 mM MgCl2, 1.75 mM CaCl2, and 0.5% Triton X-100) for the assay, but why EGTA which selectively chelate calcium divalent ion was co-excising with CaCl2? Could the authors provide the explanation for this?

  • We initially used the buffer specified in the article by Baek et al (doi: 10.1074/jbc.272.51.32042). However, in response to your feedback, we have repeated the experiment using an alternative buffer, employing GST-RBDhis (GST-fused receptor-binding domain of SARS-CoV-2) as a negative control. Our results showed no significant differences in enzyme activity between the two buffers. In the manuscript, we have now specified the Tris based buffer.

Page 5, Line 165 – Protein refolding of is the key part to obtain the functional F13 protein. Could the authors describe the procedure more in detail so that the readership can reproduce your study?

  • We have described the protein refolding procedure step by step, allowing our colleagues to reproduce our experiment.

Page 6, Figure 2 – How could the authors specify the arrow-indicated band is GST-F13 without putting the negative control (lysate from E. coli carrying empty plasmid DNA) and/or standard protein? Additionally, the authors should carry out Western blotting to quantitatively compare the relative expression levels upon the changed parameter (with/without TF and temperature).

  • We have now updated the electropherograms by adding the necessary controls. Additionally, we included Western blotting to enhance the clarity of the results.

Page 8, Figure 5 – Could the authors add the yield of GST-F13 protein (mg/L culture) in the main text?

  • We have now added data on GST-F13 protein yield to the main text of the article.

Page 8, Figure 6B – While the authors used BSA as a negative control, His-tagged GST could be the genuine negative control to evaluate the activity of the tagged F13. In addition, it would be more informative if the authors could test the inhibitory activity of NIOCH-14 (IC50).

  • As mentioned above, we have now repeated the experiment, using GST-RBDhis protein as a negative control.

In this study, we did not analyze the inhibitory activity of NIOCH-14 in details. Our work focused on the development of the producer strain. Assessing the activity of GST-F13 was necessary for protein identification. For this purpose, NIOCH-14 was used as a known inhibitor of orthopoxvirus phospholipases.

Due to the journal's specific requirements, we focused on the production and identification processes without delving deeply into the characterization of the biochemical properties of GST-F13. This characterization would indeed require conducting a series of experiments, including optimizing buffer composition and reaction conditions. Moreover, F13 may exhibit multiple enzymatic activities (such as lipase and phospholipase D activity) and interact with various substrates. Therefore, studying the effect of NIOCH-14 on F13 activity would be more appropriate using several substrates. In our view, such a scope of work and its specificity warrant a separate publication.

Page 9, Discussion – The study exclusively focused on GroEL/GroES and Trigger Factor chaperones, whereas exploring alternative or additional chaperones could be resulted in more efficient protein expressions. Also, other protein expression platform (cell-free system, Brevibacillus, yeast, insect cells, and mammalian cells) might be useful to obtain the p37 protein. Could the authors discuss the limitation of this study in the discussion section?

  • It is important to note that we made several attempts to express F13 in mammalian cells (CHO-K1 and HEK293T) as well as in the bacterium Bacillus subtilis. Unfortunately, we were unable to obtain the protein. Initially, we chose not to include this data in the manuscript, as we believed it would overly emphasize negative results.

We recognize the limitations of our study and understand the necessity of addressing this in the manuscript. Consequently, we have now expanded the discussion section to mention the potential for using other chaperones or expression systems, as well as our own experience.

The authors should provide the DNA sequence of plasmid DNA or at least amino acid sequence of GST-F13 as a supplementary information.

  • In the supplementary information, we have provided the amino acid sequence of the GST-F13 protein.

Minor Comments:

Page 1, Line 39 – “exhibits the same selectivity index (IC50) for both ...” exhibits similar half-maximal inhibitory concentration (IC50) against vaccinia virus and monkeypox virus.

Page 2, Line 90 – “…virus was obtained from the GenBank database,” Please add gene accession number.

Page 4, Line 148 – “by centrifugation at 4500 × g 15 min,” -> for 15 min,

Page 4, Line 172 – “p37-GST” -> GST-F13

Page 5, Line 205 – “VACV” means what?

Page 5, Line 212 – “After induction, a strong slowdown in the growth of the bacterial culture was observed,” If the authors can present the data, please include it. Otherwise, please add “(data not shown)”.

Page 8, Figure 6B – I suggest the authors add the number of samples (n) and the concentration of NIOCH-14 in the figure legend.

  • All the minor comments have been addressed and corrected.

We believe these changes address your concerns and enhance the overall quality of the manuscript. Thank you once again for your valuable input.

Sincerely,

 Iuliia A. Merkuleva (corresponding Author: j.a.merkulyeva@gmail.com) 

Vladimir N. Nikitin

Tatyana D. Belaya,

Egor. A. Mustaev,

Dmitriy N. Shcherbakov

Round 2

Reviewer 1 Report

Comments and Suggestions for Authors

The authors have addressed all of my concerns.It can be accepted for publication.

Author Response

Dear Reviewer,

We would like to express our sincere gratitude for the time and effort you have invested in reviewing our manuscript. Your expert opinion has been very helpful in improving our work.

Thank you for your consideration.

Sincerely,

Iuliia A. Merkuleva (corresponding Author: j,a,merkulyeva@gmail,com) 

Vladimir N. Nikitin

Tatyana D. Belaya,

Egor. A. Mustaev,

Dmitriy N. Shcherbakov

Reviewer 2 Report

Comments and Suggestions for Authors

I think the revised manuscript has been significantly improved from the initial version. Now I support the publication in the BioTech. I found some errors in the manuscript, so please consider the correction.

Page 8, Line 313 - Analysis of extracts of E. coli BL21 (DE3) cells...
italic

Page 12, Line 405 - However, despite E. coli remains...
italic

Page 14, Line 513 - paving the way for development for the development of antiviral agents
phrase overlapping

Author Response

Dear Reviewer,

We would like to express our sincere gratitude for the time and attention you have devoted to reviewing our manuscript.

Thank you for your expert opinion, which has significantly contributed to making our manuscript better. We have carefully considered your feedback and made the necessary revisions (highlited by red) to address your comments and suggestions.

Thank you for your consideration.

Sincerely,

Iuliia A. Merkuleva (corresponding Author: j,a,merkulyeva@gmail,com) 

Vladimir N. Nikitin

Tatyana D. Belaya,

Egor. A. Mustaev,

Dmitriy N. Shcherbakov